# Immune Checkpoint Inhibitors for Gastrointestinal Malignancies: An Update

**DOI:** 10.3390/cancers14174201

**Published:** 2022-08-30

**Authors:** Kathryn DeCarli, Jonathan Strosberg, Khaldoun Almhanna

**Affiliations:** 1The Lifespan Cancer Institute, The Warren Alpert Medical School of Brown University, Providence, RI 02903, USA; 2Moffitt Cancer Center, Tampa, FL 33612, USA

**Keywords:** immune checkpoint inhibitors, immunotherapy, gastrointestinal cancers

## Abstract

**Simple Summary:**

Immune checkpoint inhibitors are a class of anti-cancer therapy that work by harnessing the body’s immune system to promote cancer cell death. These drugs have become standard of care for many types of cancer, including melanoma and lung cancer, after clinical trials showed they work better than traditional chemotherapy. The role of immune checkpoint inhibitors is still evolving in the treatment of cancers of the gastrointestinal tract. This article examines the literature to support the use of immune checkpoint inhibitors to treat cancers of each part of the gastrointestinal system.

**Abstract:**

Gastrointestinal (GI) malignancies are a heterogenous group of cancers with varying epidemiology, histology, disease course, prognosis and treatment options. Immune checkpoint inhibitors (ICIs) have changed the landscape of modern cancer treatment, though they have demonstrated survival benefit in other solid tumors more readily than in GI malignancies. This review article presents an overview of the landscape of ICI use in GI malignancies and highlights recent updates in this rapidly evolving field.

## 1. Introduction

It is well-established that immune evasion plays a key role in cancer growth. Immune checkpoints such as cytotoxic T-lymphocyte-associated protein-4 (CTLA-4), programmed cell death protein-1 (PD-1), and the associated PD-1 ligand (PD-L1) have proven effective targets in the treatment of cancer. Immune checkpoint inhibitors (ICIs) aimed at these targets have changed the landscape of modern cancer treatment. Since the first approval of an ICI with ipilimumab for the treatment of metastatic melanoma in 2011 [1], ICIs have become a mainstay of antineoplastic therapy in thoracic oncology and are increasingly important in the treatment of breast and genitourinary cancers. While ICIs were initially approved in the metastatic setting, their use is now established in neoadjuvant and adjuvant settings as well. Gastrointestinal (GI) malignancies have repeatedly shown lower response rates to ICIs than other solid tumors. Approval of ICIs in the treatment of GI malignancies has lagged behind that for other primary tumors, with the exceptions of gastroesophageal cancer, hepatocellular carcinoma (HCC), and mismatch repair deficient (dMMR)/microsatellite instability-high (MSI-H) tumors [2].

GI malignancies comprise a heterogenous group of cancers with differing epidemiology, histology, clinical course, and prognosis. Classically, surgical resection has been the definitive therapy for localized GI cancers, preceded or followed by chemotherapy in high-risk populations. Biomarkers may now be used to predict response to systemic therapy with ICI, especially in the metastatic setting. Mismatch repair status has been shown to predict which patients will benefit from ICI therapy. Additionally, patients with high CPS scores might have better response to ICI therapy when combined with chemotherapy in upper GI malignancies. It is postulated that immunotherapy could be combined with other modalities to improve response rates by altering the tumor microenvironment. This review article will summarize the landscape of immunotherapy in the treatment of GI malignancies and highlight recent updates in this rapidly evolving field.

## 2. Esophageal and Gastric Cancers

An anticipated 47,000 new cases of gastroesophageal cancer will be diagnosed in the US in the year 2022 [3], and gastric cancer is the sixth most common type of cancer worldwide [4]. Squamous cell carcinoma (SCC) still accounts for the majority of gastroesophageal cancers globally. Incidence is especially high in northern Africa and eastern Asia [5]. In the US, adenocarcinoma is becoming increasingly common, often arising from pre-existing Barrett esophagus [6]. Only approximately 25% of gastric cancers are resectable at the time of diagnosis [7].

Despite the high incidence of advanced gastroesophageal cancer, chemotherapy improves survival by an average of only 6.7 months compared with best supportive care [7]. Trastuzumab [8,9] and more recently, fam-trastuzumab deruxtecan-nxki [10], have improved survival in human epidermal growth factor receptor 2 [HER2] positive disease, which accounts for 10–20% of gastroesophageal cancers [7]. However, in patients with advanced disease whose tumors are HER2 negative, effective treatment options remain limited. In the early 2000s, epirubicin, cisplatin and continuous-infusion fluorouracil (ECF) was the standard of care (SOC) systemic therapy. Leucovorin, 5-fluorouracil, and oxaliplatin (FOLFOX) was then shown to reduce toxicity and trend towards improved progression free survival (PFS) and overall survival (OS) [11]. FOLFOX displaced ECF as the standard of care in 2016 based on results of the phase II CALGB 80403 trial, showing similar efficacy with reduced toxicity of FOLFOX [12]. However, the median OS for advanced gastroesophageal cancer remains less than one year.

PD-L1 expression is detected in approximately 50% of esophageal squamous cell carcinomas [13] and up to 60% of gastric cancer specimens [14], suggesting a role for PD-1 inhibitors. The response rate in gastroesophageal cancers to ICI monotherapy ranges from 5 to 30% [15]. This may be due in part to the heterogeneity seen in gastroesophageal cancers. Table 1 summarizes landmark trials of ICIs in gastroesophageal cancers. 

Pembrolizumab was approved in the US in 2017 for third-line use in patients with metastatic disease and CPS score ≥1 [16] in the wake of positive findings from the phase II KEYNOTE 059 trial. In this study, 57.1% of the study population had PD-L1 positive tumors, and the objective response rate was 15.5% among PD-L1 positive tumors versus 6.4% among PD-L1 negative tumors [15]. In a negative study, the phase III KEYNOTE 061 trial did not show superiority of pembrolizumab over paclitaxel as second line treatment in all-comers or in patients with CPS score ≥1, who comprised 66.7% of the study population [17]. However, in a 2-year update of this study, a post hoc analysis found improved 24 month OS in patients with PD-L1 rich tumors when the study population was stratified by PD-L1 expression into groups with CPS ≥1, CPS ≥5, and CPS ≥10 [18]. Subsequently, the phase III KEYNOTE 062 trial showed non-inferiority of pembrolizumab versus chemotherapy in the first line setting for patients with CPS score ≥1, with a more tolerable safety profile [19]. That trial included only patients with CPS score of one or more. More recently, the addition of first line pembrolizumab to SOC chemotherapy was associated with improved OS for all comers and especially for patients with SCC and CPS ≥10 in the KEYNOTE 590 trial, in which 51% of enrolled patients had PD-L1 CPS of 10 or more [20]. Overall survival (OS) was also improved with pembrolizumab monotherapy over chemotherapy in this patient population in the phase III KEYNOTE 181 trial, with 35.4% of enrolled patients having CPS score of 10 or more [21]. The phase III KEYNOTE 811 trial, still ongoing at time of this writing, has shown improved objective response rate (ORR) in patients with metastatic HER2 + disease treated with pembrolizumab plus standard of care (SOC) trastuzumab and chemotherapy versus SOC alone in the first interim analysis [9]. The ongoing phase III KEYNOTE 859 trial is examining pembrolizumab in the first line setting to treat HER2 negative disease [22].

As in other solid tumors, pembrolizumab has gained acceptance first in the third line metastatic setting and now shows promise for use in earlier phases of treatment. Data presented at ASCO 2021 suggest promise of neoadjuvant chemotherapy and radiation plus pembrolizumab in association with higher rates of major pathologic response [MPR] at time of surgery [23] for patients with esophageal adenocarcinoma.

Nivolumab has also evolved as an important tool in the treatment of gastroesophageal cancer, including resectable disease. The ATTRACTION 2 and ATTRACTION 4 trials showed nivolumab improved outcomes among Asian patients with metastatic disease in both the chemotherapy-refractory and first line settings [24,25]. In 2018, the phase II CheckMate 032 trial established clinical activity of nivolumab as second line treatment for advanced disease regardless of CPS score in a Western population [26]. This study also introduced clinical activity of combination nivolumab/ipilimumab. The phase III Checkmate 577 trial investigated adjuvant nivolumab after neoadjuvant chemoradiation followed by surgical resection of esophageal or gastroesophageal junction (GEJ) cancer. Compared with placebo, adjuvant nivolumab significantly improved disease-free survival (DFS) [27].

The phase III Checkmate 649 trial, the first global study of nivolumab in the first line setting, also showed improved OS and progression free survival (PFS) in patients with metastatic disease who received first line nivolumab + chemotherapy versus chemotherapy alone, regardless of CPS score [28]. This effect was seen in all PD-L1 expression groups and across multiple pre-specified subgroups including microsatellite unstable disease. Notably, this study marked the first time that OS surpassed one year among patients with metastatic HER2 negative gastroesophageal cancer in the first line treatment setting. Nivolumab was granted accelerated approval in 2021 in response to these findings for treatment in combination with first line chemotherapy [29].

More recently, the phase III Checkmate 648 trial showed improved median OS among patients with metastatic SCC of the esophagus treated with first-line nivolumab plus chemotherapy or nivolumab plus ipilimumab compared with chemotherapy alone; this effect was seen both in patients with tumors expressing PD-L1 > 1% and in the overall study population [30]. This practice-changing study introduces first-line nivolumab as standard of care in the treatment of metastatic esophageal SCC, in combination with either fluoropyrimidine- and platinum-based chemotherapy or ipilimumab for patients who are not candidates for chemotherapy [31]. These regimens were approved by FDA in May 2022 [32].

Avelumab and camrelizumab have shown less promise. The phase III JAVELIN 100 study showed avelumab maintenance after first line chemotherapy did not result in superior OS vs. continued chemotherapy in the primary population of all randomly assigned patients with advanced gastric and GEJ cancer, nor in patients with tumor cell PD-L1 expression of 1% or higher [33]. As third line treatment, avelumab also did not improve OS or PFS compared to chemotherapy in the phase III JAVELIN 300 study [34]. Phase II studies of camrelizumab combined with apatinib suggest anti-tumor activity among patients with advanced gastric or GEJ adenocarcinoma [35,36].
cancers-14-04201-t001_Table 1Table 1Landmark trials of immunotherapy in esophageal and gastric cancers.TrialYearTrialDesignLocationStudy ArmsPatient PopulationNOutcomeATTRACTION 2 [24]2017Phase III, randomized, double blindJapan, South Korea, TaiwanNivolumab monotherapy vs. placeboAdvanced disease, progressed after second line therapy493Improved median OS in nivolumab arm vs. placebo arm (5.26 months vs. 4.14 months, HR 0.63, 95% CI 0.51–0.78, *p* < 0.0001)KEYNOTE 059 [15]2018Phase II, open labelGlobal, 17 countriesPembrolizumab monotherapyAdvanced disease, progressed after second line therapy259ORR 11.6%, CR 2.3%, MDR 8.4 months, 17.8% of pts experienced grade 3–5 treatment related adverse eventsCheckMate 032 [26]2018Phase II, open labelUS and 5 European countriesNivolumab monotherapy vs. nivolumab + ipilimumab (low dose) vs. nivolumab + ipilimumab (high dose)Advanced disease, progressed after first line or subsequent therapy160ORR 12% (95% CI, 5% to 23%) in nivolumab arm, 24% (95% CI, 13% to 39%) in NIVO/IPI1 arm, 8% in NIVO/IPI3 arm (95% CI, 2% to 19%)KEYNOTE 061 [17]2018Phase III, randomized, open labelGlobal, 30 countriesPembrolizumab monotherapy vs. paclitaxelAdvanced disease, progressed after first line therapy592 (395 with CPS ≥ 1)Improved median OS in pembrolizumab arm vs. paclitaxel arm for pts with CPS ≥ 1 (9.1 vs. 8.3 months, HR 0.82, 95% CI 0.66–1.03, one-sided *p =* 0.0421)ATTRACTION 4 [25]2019Phase II–III, randomized, double blindJapan, South Korea, TaiwanNivolumab + chemotherapy vs. placebo + chemotherapyPreviously untreated, HER2 negative, unresectable disease724Improved median PFS in nivolumab arm (10.45 vs. 8.34 months, HR 0.68, 98.51% CI 0.51–0.90, *p =* 0.0007), no significant difference in OSKEYNOTE 590 [20]2019Phase III, randomized, controlled, double blindedGlobal, 26 countriesPembrolizumab + chemotherapy vs. chemotherapy alonePreviously untreated, advanced esophageal or GEJ cancer (mainly SCC)749Improved median OS in pembrolizumab + chemo arm vs. chemo alone arm for all pts (12.4 vs. 9.8 months, *p* < 0.0001) and for SCC histology with CPS ≥10 (13.9 vs. 8.8 months, *p* < 0.001)KEYNOTE 062 [19]2020Phase III, randomized, controlled, partly blindedGlobal, 29 countriesPembrolizumab vs. pembrolizumab + chemotherapy vs. chemotherapy alonePreviously untreated, advanced disease, CPS ≥1763Pembrolizumab non-inferior to chemotherapy with improved safety profile. Improved median OS in pembrolizumab arm vs. chemotherapy alone arm for pts with CPS ≥10 (17.4 vs. 10.8 months, HR 0.69, 95% CI 0.49–0.97, not statistically tested).KEYNOTE 181 [21]2020Phase III, randomized, open labelGlobal, 32 countriesPembrolizumab monotherapy vs. chemotherapy aloneAdvanced disease, progressed after first line therapy628Improved OS in pembrolizumab arm vs. chemotherapy arm for patients with CPS ≥10 (9.3 vs. 6.7 months, HR 0.69, 95% CI 0.52–0.93, *p =* 0.0074); effect seen only in SCC, not in adenocarcinomaCheckMate 577 [27]2021Phase III, randomized, double blindGlobal, 29 countriesNivolumab vs. placebo as adjuvant treatmentEsophageal or GEJ cancer status post neoadjuvant chemoradiation and resection1085Improved median DFS in nivolumab arm vs. placebo arm (22.4 months vs. 11.0 months, HR 0.69, 96.4% CI 0.56–0.86, *p* < 0.001)CheckMate 649 [28]2021Phase III, randomized, open labelGlobal, 29 countriesNivolumab + chemotherapy vs. nivolumab + ipilimumab vs. chemotherapy alonePreviously untreated, unresectable, HER2 negative1581Improved median OS in nivo + chemo arm vs. chemo alone arm in CPS ≥5 group (14.4 vs. 11.1 months, HR 0.71, 98.4% CI 0.59–0.86, *p* < 0.0001), CPS ≥1 group (14.0 vs. 11.3 months, HR 0.77, 99.3% CI 0.64–0.92, *p* < 0.0001) and overall population (13.8 vs. 11.6 months, HR 0.80, 99.3% CI 0.68–0.94, *p =* 0.0002)KEYNOTE 811 [9]2021Phase III, randomized, double blindGlobal, 20 countiresPembrolizumab + trastuzumab + chemotherapy vs trastuzumab + chemotherapyPreviously untreated HER2+ metastatic disease264 analyzed in first interim analysisImproved ORR in pembrolizumab + SOC arm vs. placebo + SOC arm (74.4% vs. 51.9%, difference 22.7 percentage points, 95% CI 11.2–33.7, *p =* 0.00006]ASCO [23]2021Phase II, randomized, open labelUSNeoadjuvant pembrolizumab + chemoradiotherapy vs. neoadjuvant chemoradiotherapy alonePreviously untreated, resectable disease, eligible for curative surgery31 analyzed to dateImproved MPR rate (50.0%) compared to historical dataCheckmate 648 [30]2022Phase III, randomized, open labelGlobal, 26 countriesNivolumab + chemotherapy vs. nivolumab + ipilimumab vs. chemotherapy alonePreviously untreated, unresectable advanced, recurrent or metastatic esophageal squamous cell carcinoma970Improved median OS in nivo + chemo arm vs. chemo alone arm both in PD-L1 >1% group (15.4 vs. 9.1 months, HR 0.54, 99.5% CI 0.37–0.80, *p* < 0.001) and overall population (13.2 vs. 10.7 months, HR 0.74, 99.1% CI 0.59–0.96, *p =* 0.002) Improved median OS in nivo + ipi arm vs. chemo alone arm in PD-L1 >1% group (13.7 vs. 9.1 months, HR 0.64, 98.6% CI 0.46–0.90, *p =* 0.001) and overall population (12.7 vs. 10.7 months, HR 0.78, 98.2% CI 0.62–0.98, *p =* 0.01)KEYNOTE 859 [22]Not yet publishedPhase III, randomized, double blindGlobal, 33 countriesPembrolizumab + chemotherapy vs. placebo + chemotherapyPreviously untreated, unresectable, HER2 negativeN/AForthcoming

## 3. Colorectal Cancer

From birth to death, there is a 1 in 24 chance for men and a 1 in 25 chance for women to develop colorectal cancer in the US, and it is the second leading cause of cancer death [3]. All dMMR/MSI-H tumors account for approximately 15% of colorectal cancers and show an increased rate of response to treatment with ICIs [37]. Table 2 summarizes landmark trials of immunotherapy in the treatment of dMMR/MSI-H colorectal cancer.

Pembrolizumab is now standard of care first-line treatment for patients with dMMR/MSI-H tumors based on data from the phase III KEYNOTE-177 study showing superior median PFS versus chemotherapy (16.5 vs. 8.2 months, HR 0.60, 95% CI 0.45–0.80, *p =* 0.0002), with fewer treatment-related adverse events [37]. The results of the phase II CheckMate 142 study led to FDA approval of nivolumab given alone or in combination with ipilimumab as second-line treatment for MSI-H/dMMR metastatic disease [38]. The first-line therapy cohort from this study also derived clinical benefit from combination nivolumab/ipilimumab, with ORR 69% (95% CI, 53–82%) and disease control rate 84% (95% CI, 70.5–93.5) [39]. Adjuvant trials are ongoing.

Unfortunately, microsatellite stable (MSS) disease, which accounts for the vast majority of colorectal cancer, has shown no clear increased response to immunotherapy in several trials of atezolizumab [40,41] and tremelimumab plus durvalumab [42]. Mutations in the *POLE-1* gene have been associated with high levels of neoantigens and tumor-infiltrating lymphocytes (TILs) in both the tumor and the tumor microenvironment (TME), suggesting *POLE-1* mutational status in MSS metastatic colorectal cancer could be used to predict good response to ICI therapy [43].

A recent phase II study of the anti-PD1 monoclonal antibody dostarlimab1 in 12 patients with locally advanced MSI-H rectal cancer showed 100% clinical CR [44]. This promising finding will require further study in the phase III setting.
cancers-14-04201-t002_Table 2Table 2Landmark trials of immunotherapy in colorectal cancers.TrialYearTrialDesignLocationStudy ArmsPatient PopulationNOutcomeCheckMate 142 [38]2017Phase II, open label8 countriesNivolumab monotherapyMetastatic dMMR/MSI-H disease, progressed on at least 1 prior line of treatment74ORR 31.1% (95% CI 20.8–42.9%) at median follow up of 12 monthsKEYNOTE 177 [37]2020Phase III, randomized, open label23 countriesPembrolizumab vs. 5-FU-based chemotherapy +/− bevacizumab or cetuximabMetastatic dMMR/MSI-H disease, no prior systemic therapy307Improved median PFS in pembrolizumab arm vs. chemotherapy arm (16.5 vs. 8.2 months, HR 0.60, 95% CI 0.45–0.80, *p =* 0.0002)CheckMate 142 [39]2022Phase II, open label6 countriesNivolumab + ipilimumabMetastatic dMMR/MSI-H disease, no prior systemic therapy45ORR 69% (95% CI 53–82%) at median follow up of 29 months

## 4. Hepatocellular Carcinoma

Hepatocellular carcinoma (HCC) is the most common type of liver cancer and constitutes the second most common cause of global cancer mortality. Incidence is especially high in Asia, and most cases are due to chronic hepatitis B or C infection [45]. Advanced and metastatic disease carry poor prognosis. Treatment algorithms take Child–Pugh score into account; systemic therapy is given for extrahepatic metastases and for intrahepatic disease not amenable to locoregional treatment in patients with Child–Pugh score A or low B. The small-molecule multikinase vascular endothelial growth factor receptor (VEGFR) inhibitor sorafenib became the standard of care for systemic therapy of advanced HCC in 2007 based on results of the phase III SHARP trial showing a median OS benefit of 10.7 vs. 7.9 months [46]. This treatment can be challenging for patients to tolerate due to hand-foot syndrome and other cutaneous toxicities. Regorafenib and lenvatinib are also approved for front-line treatment of metastatic HCC, with similar efficacy and side effect profiles. Efforts have been underway to identify applications for immunotherapy, thought promising in HCC due to its high immunogenicity [47]. Table 3 summarizes landmark trials of immunotherapy in the treatment of HCC.

Results of the phase I/II CheckMate 040 study led to accelerated approval of nivolumab in the treatment of HCC in 2017. This trial showed clinical benefit in the second-line setting after disease progression or treatment-limiting side effects on sorafenib, with a median OS of 15 months and RR 15% [48]. Another cohort from this study was randomized to nivolumab plus cabozantinib versus nivolumab plus ipilimumab plus cabozantinib; ORR was improved in the triple therapy arm (26% vs. 17%) [49]. Clinical activity was also seen in patients treated with combined ipilimumab plus nivolumab [50], leading to accelerated FDA approval for this regimen in 2020 [51]. The recently published phase III CheckMate 459 trial was the first study to evaluate nivolumab compared with sorafenib in the first-line setting; though nivolumab was well tolerated and trended towards improved median OS, this finding did not reach statistical significance [52]. As a result, single agent nivolumab may now be considered as a first-line systemic therapy option only for patients with Child–Pugh score A or B who are ineligible for sorafenib [52].

Atezolizumab combined with bevacizumab became the standard of care for advanced HCC based on the results of the phase III IM-brave150 trial [53]. In this study, 501 treatment naïve patients with advanced HCC were randomized to receive atezolizumab plus bevacizumab versus then standard of care sorafenib. The investigators found an improved HR for death in the atezolizumab plus bevacizumab arm compared with the sorafenib arm (HR 0.58, 95% CI 0.42–0.79, *p* < 0.001). Improved median OS was also seen at 12 months (67.2%, 95% CI 61.3–73.1 vs. 54.6%, 95% CI 45.2–64.0) [53]. The FDA approved atezolizumab plus bevacizumab as front-line therapy for unresectable HCC in May 2020 [54].

Pembrolizumab was approved as second-line therapy after it showed clinical activity in the KEYNOTE 224 trial in 2018 [55]. However, the subsequent KEYNOTE 240 trial did not meet primary endpoints in OS and PFS [56]. The phase III KEYNOTE 394 trial showed statistically significant increase in median OS among Asian patients with advanced HCC treated with pembrolizumab in the second-line setting vs. placebo [57], leading to FDA accelerated approval in patients previously treated with sorafenib [58]. The ongoing phase II PRIMER-1 study is investigating pembrolizumab plus lenvatinib in resectable HCC based on promising results from a phase Ib study [59].

Finally, the phase III HIMALAYA trial shows that compared with sorafenib, the combination of durvalumab and tremelimumab improves OS in patients with unresectable HCC [60].

While the regimens discussed above have not all been compared head to head, multiple immunotherapy treatment options are now available to patients with metastatic HCC. The National Comprehensive Cancer Network (NCCN) prefers first line systemic therapy with atezolizumab combined with bevacizumab for patients with Child–Pugh class A cirrhosis who do not have contraindications such as risk for bleeding from esophageal varices. Pembrolizumab, nivolumab, and durvalumab are also acceptable as first-line systemic therapy. Data is lacking to support the use of immunotherapy in the second-line setting for patients who received first-line immunotherapy. However, combined nivolumab plus ipilimumab is recommended as second-line therapy for patients who previously received sorafenib. Pembrolizumab, single agent nivolumab, and dostarlimab maintain a category 2B recommendation in this setting [61].
cancers-14-04201-t003_Table 3Table 3Landmark trials of immunotherapy in hepatocellular cancers.TrialYearTrialDesignLocationStudy ArmsPatient PopulationNOutcomeCheckMate 040 [48]2017Phase I/II, open-label, non-comparative, dose escalation and expansion trial11 countriesNivolumab monotherapyAdvanced disease, progressed on or unable to tolerate sorafenib262 (28 dose-escalation, 214 dose-expansion)ORR 15% (95% CI 6–28%) in dose escalation phase; 20% (95% CI 15–26) in dose expansion phaseCheckMate 040 [49]2020Nivolumab 1 mg/kg plus ipilimumab 3 mg/kg Q3 weeks × 4 followed by nivolumab 240 mg Q2 weeks (A) vs. nivolumab 3 mg/kg plus ipilimumab 1 mg/kg Q3 weeks × 4 followed by nivolumab 240 mg Q2 weeks (B) vs. nivolumab nivolumab 3 mg/kg Q2 weeks plus ipilimumab 1 mg/kg Q6 weeks (C)148ORR 32% (95% CI 20–47%) in arm A, 27% (95% CI 15–41%) in arm B, 29% (95% CI 17–43%) in arm CCheckMate 040 [50]2020Nivolumab plus cabozantinib vs. nivolumab plus ipilimumab plus cabozantinib71ORR 17% in nivolumab plus cabozantinib arm; ORR 26% in nivolumab plus ipilimumab plus cabozantinib armKEYNOTE 224 [55]2018Phase II, non-randomized10 countriesPembrolizumab monotherapyAdvanced disease, progressed on or unable to tolerate sorafenib104ORR 17% (95% CI 11–26%)KEYNOTE 240 [56]2020Phase III, randomized, double blind27 countriesPembrolizumab vs. placeboAdvanced disease, progressed on first line sorafenib413Did not meet primary endpoints in OS and PFSIMbrave150 [53]2020Phase III, randomized, open label17 countriesAtezolizumab plus bevacizumab vs. sorafenibUnresectable disease with no previous systemic treatment501Improved HR for death in atezolizumab-bevacizumab arm vs. sorafenib arm (0.58, 95% CI 0.42–0.79, *p* < 0.001); improved OS at 12 months (67.2%, 95% CI 61.3–73.1 vs. 54.6%, 95% CI 45.2–64.0)CheckMate 459 [52]2022Phase III, randomized, open label22 countriesNiviolumab monotherapy vs. sorafenib monotherapyAdvanced disease with no previous systemic treatment743Improved median OS in nivolumab vs. sorafenib group (16.4 vs. 14.7 months, HR 0.85, 95% CI 0.72–1.02, *p =* 0.075), not statistically significantKEYNOTE 394 [57]2022Phase III, randomized, double blindAsiaPembrolizumab vs. placeboAdvanced disease, progressed on first line therapy453Improved OS in pembrolizumab vs. placebo group (14.6 vs. 13.0 months, HR 0.79, 95% CI 0.63–0.99, *p =* 0.018)HIMALAYA [60]2022Phase III, randomized, open-label16 countriesTremelimumab plus durvalumab vs. sorafenib vs. durvalumabUnresectable disease, no prior systemic therapy1171Improved 3 year OS in tremelimumab plus durvalumab vs. sorafenib group (30.7% vs. 20.2%, HR 0.78, 96% CI 0.65–0.92, *p =* 0.0035)

## 5. Cholangiocarcinoma

Unresectable or metastatic biliary tract cancers carry a poor prognosis with median OS on the order of one year [62]. The primary treatment for these cancers was established in 2010 as gemcitabine plus cisplatin [63]. In 2020, a phase II trial of 54 patients showed clinical activity of nivolumab in heavily pre-treated advanced hepatobiliary cancer, with 22% partial response and 37% stable disease. All patients who showed a partial response had MMR proficient (pMMR) disease, and 90% had disease that expressed levels of PD-L1 ≥1%. Median OS was 14.24 months and median PFS was 3.68 months [64]. In the KEYNOTE-158 and KEYNOTE-028 studies, pembrolizumab resulted in objective response in 6–13% of patients with advanced biliary tract cancer [62]. Improved biomarkers are needed to guide patient selection for immunotherapy as second-line treatment for refractory advanced cholangiocarcinoma.

Based on the results of the phase III TOPAZ trial, durvalumab was approved in 2022 as first-line therapy in addition to standard of care gemcitabine plus cisplatin [65]. In this study, 685 patients with previously untreated unresectable locally advanced, recurrent or metastatic biliary tract cancer were randomized to durvalumab versus placebo in combination with gemcitabine and cisplatin. The durvalumab plus chemotherapy arm experienced significantly improved OS (HR 0.80, 95% CI 0.66–0.97, *p =* 0.021). While median OS only improved by 1.3 months (12.8 versus 11.5 months), over twice as many subjects were alive at two years in the experimental arm (24.9% vs. 10%). In this study, 55% of patients had intrahepatic cholangiocarcinomas, 19% had extrahepatic cholangiocarcinomas, and 25% had gallbladder cancer [66].

As in other noncolorectal cancers, dMMR/MSI-H tumors can be treated with pembrolizumab, though only 22 patients with advanced hepatobiliary cancer were enrolled in the phase II KEYNOTE-158 trial [67]. Only about 10% of patients with cholangiocarcinomas harbor dMMR/MSI-H disease [2].

## 6. Anal Cancer

Anal cancer is unique among GI malignancies in that it is less prevalent than other primary sites, comprises mainly squamous cell histology, is highly associated with human papillomavirus (HPV), and commonly presents with localized or locally advanced disease [68]. In the metastatic setting, the standard of care for first-line systemic therapy is carboplatin given concurrently with paclitaxel based on the InterAACT trial [69]. A phase II study showed clinical activity and safety of single-agent nivolumab in treatment-refractory metastatic SCC of the anal canal. In this single-arm trial performed at 10 academic centers in the United States, 37 patients received at least one dose of nivolumab, with an ORR of 24% [95% CI 15–33%] and no serious adverse events reported [70]. The phase Ib KEYNOTE-028 trial, which enrolled 25 patients with PD-L1 positive advanced anal carcinoma, showed ORR 17% (95% CI 5–37%) and tolerable safety profile [71]. NCCN guidelines now prefer nivolumab or pembrolizumab as subsequent therapy for patients with recurrent disease who have not previously received immunotherapy [72].

## 7. Pancreatic Cancer

Front-line treatment for metastatic or unresectable pancreatic cancer is cytotoxic chemotherapy with 5-fluorouracil, leucovorin, irinotecan, and oxaliplatin (FOLFIRINOX) or gemcitabine plus nab-paclitaxel. Pancreatic cancer has been seen as a non-immunogenic malignancy due to an immunosuppressive tumor microenvironment [73,74,75]. ICIs have shown no clinical benefit in the treatment of pancreatic cancer. For example, a phase II trial of single-agent ipilimumab showed no clinical activity [76]. Combining ICIs with traditional chemotherapy has not yielded promising results in several phase I trials [77,78]. Dual immune checkpoint blockade with durvalumab and tremelimumab has also shown disappointing results in a phase II trial [79]. Similarly, pancreatic cancer vaccines to increase tumor immunogenicity have showed limited clinical benefit [80,81,82]. dMMR/MSI-H disease accounts for only around 2% of pancreatic cancers [83], making second-line pembrolizumab available to a small minority of patients. A phase I trial of mogamulizumab, an anti-CC chemokine receptor 4 antibody, combined with either durvalumab or tremelimumab enrolled only three patients with pancreatic cancer and showed no clinical activity [84]. Several ongoing trials are examining chimeric antigen receptor (CAR)-T cell therapy and other novel agents in this disease. A recent case report described a patient with progressive metastatic pancreatic cancer who experienced a 72% response by Response Evaluation Criteria in Solid Tumors after receiving autologous T cells genetically engineered to target the KRAS G12D mutation [85].

## 8. Small Bowel Cancer

Front-line treatment for advanced small bowel adenocarcinoma is similar to that for other GI malignancies with a fluoropyrimidine-based cytotoxic chemotherapy regimen. There is no approval for ICIs in the treatment of small bowel cancer. Due to the low incidence of small bowel cancer, there is a small population of patients on which to build an evidentiary base. A phase II study found no clinical activity of pembrolizumab in treatment of pMMR/MSS small bowel adenocarcinoma; however high tumor mutational burden (TMB) was associated with an increased response rate [86].

## 9. Neuroendocrine Neoplasms

Neuroendocrine neoplasms are highly diverse malignancies which are broadly categorized into well-differentiated neuroendocrine tumors (NET) and poorly differentiated neuroendocrine carcinomas (NEC). These entities are clinically and biologically distinct: NETs are characterized by a relatively slow growth rate and very low TMB, whereas poorly differentiated NECs are highly aggressive, with a higher TMB, on average [87,88]. ICIs have generally shown low rates of activity in well-differentiated NETs: a phase II study of pembrolizumab was associated with an overall response rate of only 3.7% [89]. Poorly differentiated NECs appear to be somewhat more responsive to immunotherapy, particularly with combination of CTLA-4 and PD-1/L1 inhibitors. A basket study of ipilimumab plus nivolumab in a heterogeneous cohort of patients with neuroendocrine neoplasms demonstrated responses only in the high-grade cohort which consisted primarily of poorly differentiated NEC [90]. An expansion cohort of 19 patients in this population demonstrated ORR of 26% [91]. A larger retrospective study of 34 patients with high grade tumors (79% NEC) who had progressed on at least one line of cytotoxic chemotherapy demonstrated an ORR of 14.7% [92]. Median OS was only five months. In summary, immunotherapy currently plays a relatively minor role in neuroendocrine neoplasms, but can be considered in platinum-refractory NECs where options are limited. In this scenario, combination treatment using ipilimumab/nivolumab is typically recommended.

## 10. Conclusions and Future Directions

GI malignancies remain a heterogenous group of cancers with wide variability in disease course, prognosis, and tumor microenvironment. Predictive biomarkers could be used in the future to select which patients stand to benefit the most from ICIs for GI malignancies. Similarly, they could be used to support de-escalation of treatment and avoid unnecessary toxicity. PD-L1 expression and dMMR/MSI-H status are examples of such biomarkers, but these have shown specific and narrow clinical applications. Identification of new clinically applicable diagnostic and prognostic biomarkers should be an area of priority in future research. Cytolytic T cell activity and immune activation indices are examples of promising measures and signals for future drug development [93]. Resistance to ICIs must also be addressed. To this end, techniques under current investigation include combination therapy to alter the tumor microenvironment and target mechanisms of immune evasion [94].

While trials prior to 2019 focused mainly on IO monotherapy, the past few years have seen the fruition of trials comparing combination therapy with standard of care chemotherapy alone. The addition of IO to cytotoxic chemotherapy regimens is now approved as first-line treatment for multiple GI malignancies, representing a possible future trend. Next steps may include further comparison of monotherapy versus combination therapy. We also anticipate emphasis in future research on the role of ICIs in the adjuvant, neoadjuvant, and maintenance settings.

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
