# Peer review of "Immune Checkpoint Inhibitors for Gastrointestinal Malignancies: An Update"

_cancers, 2022, doi:10.3390/cancers14174201_

Round 1
Reviewer 1 Report
The manuscript “Immune checkpoint inhibitors for gastrointestinal malignancies: an update” is a timely review of ICI for GI cancers. Overall, this is a high-quality article with recently published information.
Two major comments:
1. There should be a table for colorectal cancer clinical trials.
2. When possible, it is important to include molecular pathology information about the cancers in each clinical trial. For example, are the tumors PD-L1 positive or not?
I suggest acceptance after minor revisions.
Author Response
Thank you very much for this helpful feedback. Please see response to each point below. We hope this adequately addresses your concerns and welcome additional comments.
Sincerely,
Kathryn DeCarli on behalf of the author team
- There should be a table for colorectal cancer clinical trials.
We have added Table 2, including 3 studies (pembro, nivo, nivo+ipi) for dMMR/MSI-H colorectal cancer.
2. When possible, it is important to include molecular pathology information about the cancers in each clinical trial. For example, are the tumors PD-L1 positive or not?
Not all trials clearly report molecular pathology information, and some are exploratory analyses, so it is challenging to standardize this in a table column. We have added data to the text in section 2 (lines 70-86) about CPS score in metastatic gastroesophageal cancers. Section 3 also addresses colorectal dMMR/MSI-H disease.
Reviewer 2 Report
This review paper by DeCarli et al nicely summarizes recent progress in the area of checkpoint blockade therapy for GI malignancies. I recommend publishing the review after addressing the points below:
1. The review lacks key illustrative figures for the topic.
2. Table 1 contains unnecessary information, such as author name and journal name. Removal of this information will make it easier for readers to digest the information.
3. The authors highlighted combination therapy frequently under each cancer type. I believe the addition of a separate paragraph that summarizes the current status of combination therapy in comparison to monotherapies and future directions will be useful.
Author Response
Thank you very much for this helpful feedback. Please see response to each point below. We hope this adequately addresses your concerns and welcome additional comments.
Sincerely,
Kathryn DeCarli on behalf of the author team
1. The review lacks key illustrative figures for the topic.
We feel that the tables are helpful visual aids to present the trials discussed in the article, especially since we address multiple malignancies. We have added a third table based on Reviewer 1's comments. A figure illustrating T cell activation may be redundant as the mechanisms of action of immunotherapy drugs are well known to the article's target audience, and are not the subject of this article.
2. Table 1 contains unnecessary information, such as author name and journal name. Removal of this information will make it easier for readers to digest the information.
We have removed author and journal name from all tables.
3. The authors highlighted combination therapy frequently under each cancer type. I believe the addition of a separate paragraph that summarizes the current status of combination therapy in comparison to monotherapies and future directions will be useful.
We have added the following text in lines 331-335: "While trials prior to 2019 focused mainly on IO monotherapy, the past few years have seen the fruition of trials comparing combination therapy with standard of care chemotherapy alone. The addition of IO to cytotoxic chemotherapy regimens is now approved as first line treatment for multiple GI malignancies, representing a possible future trend. Next steps may include further comparison of monotherapy versus combination therapy."